# Tepary Bean (*Phaseolus acutifolius*) Lectins as Modulators of Intracellular Calcium Mobilization in Breast Cancer and Normal Breast Cells

**DOI:** 10.3390/ijms26031064

**Published:** 2025-01-26

**Authors:** Andrea Díaz-Betancourt, María Elizabeth Galicia-Castillo, Verónica Morales-Tlalpan, Jorge Luis Chávez-Servín, Alejandro Blanco-Labra, Teresa García-Gasca, Carlos Saldaña

**Affiliations:** 1Laboratorio de Biofísica de Membranas y Nanotecnología, Facultad de Ciencias Naturales, Universidad Autonoma de Queretaro, Av. De las Ciencias s/n, Juriquilla, Queretaro 76230, Queretaro, Mexico; andrea.dzb@hotmail.com (A.D.-B.); mariagalicia2509@gmail.com (M.E.G.-C.); vmtlalpan@gmail.com (V.M.-T.); 2Laboratorio Nacional de Visualización Científica Avanzada, Facultad de Ciencias Naturales, Universidad Autonoma de Queretaro, Av. De las Ciencias s/n, Juriquilla, Queretaro 76230, Queretaro, Mexico; 3Laboratorio de Biología Celular y Molecular, Facultad de Ciencias Naturales, Universidad Autonoma de Queretaro, Av. De las Ciencias s/n, Juriquilla, Queretaro 76230, Queretaro, Mexico; jorge.chavez@uaq.mx; 4Departamento de Biotecnología y Bioquímica, Centro de Investigación y Estudios Avanzados del IPN, Km. 9.6 Libramiento Norte, Carretera Irapuato-León, Irapuato 36824, Guanajuato, Mexico; alejandroblancolabra@gmail.com

**Keywords:** breast cancer cells, intracellular calcium, lectins, *Phaseolus acutifolius*, Tepary bean, TBFL, rTBL-1

## Abstract

Lectins are proteins that specifically recognize carbohydrates on cell membranes, triggering several cellular events such as apoptosis of cancer-transformed cells; however, the mechanisms of action remain incompletely understood. Our research group has reported that a concentrated fraction of Tepary bean lectins (*Phaseolus acutifolius*; TBLF) exhibits the concentration-dependent induction of apoptosis in colon cancer cells by caspase activation. It is well established that an increase in cytoplasmic calcium ([Ca^2+^]_i_) initiates intracellular signals involved in processes such as exocytosis, gene transcription, apoptosis, cell cycle regulation, and muscle contraction, among others. Furthermore, dysregulated calcium signaling has been implicated in various diseases, including certain neurological disorders and cancer. In this study, we aim to demonstrate the effects of native TBLF lectins and a recombinant lectin (rTBL-1) on calcium mobility in breast cancer cells (MCF-7) and non-cancerous cells (MCF-12F). Both TBLF and rTBL-1 increased intracellular calcium concentrations and mobilized calcium from intracellular stores in a concentration-dependent manner; however, the two cell lines exhibited differential responses. While MCF-12F cells restored cytoplasmic calcium concentration, MCF-7 cells maintained a high intracellular calcium concentration. This strongly suggests that lectins can elicit differential cellular responses in cancer and non-cancer cells due to variations in their intrinsic mechanisms of calcium homeostasis. Finally, we demonstrated that TBLF and rTBL-1 can differentially alter Metabolic Cellular Activity (MCA) as a direct measure of cell viability (CVi) in both cell lines. These findings strengthen the evidence of the therapeutic potential of Tepary bean lectins. Undoubtedly, further studies will be necessary to elucidate the mechanisms underlying their applications.

## 1. Introduction

Legume lectins are the most studied family of plant lectins; they are mainly proteins of 25 to 30 kDa with binding sites for the metal ions Ca^2+^, Mn^2+^, and Mg^2+^ [1]. The functional protein is composed of dimers or tetramers of identical subunits [1,2]. These proteins have been extensively studied for biomedical purposes because of the recognition properties of cell membrane glycans [3], including cancer diagnosis and treatment [4,5,6,7,8]. A Tepary bean (*Phaseolus acutifolius*) lectin fraction (TBLF) has been studied by our working group due to its cytotoxic effects on cancer cell lines of different lineages such as breasts, the cervix, and the colon, where MCF-7 breast cancer cells and CaCo2 colon cancer cells were the most sensitive to TBLF [7], and several colon cancer cell lines have shown differential cytotoxic effects, apoptosis induction, and cell cycle arrest [9]. Apoptosis has been related to caspase activity and p53 phosphorylated in serine 46 [6,9]. TBLF also exhibits early tumorigenesis inhibition in rats when they were subjected to chemically induced cancer by apoptosis induction [10].

As TBLF requires a long and costly process, we focused on producing a recombinant lectin (rTBL-1) in *Pichia pastoris* yeast [11,12], which exhibits similar cytotoxic effects to TBLF, where interaction with epidermal growth factor receptor (EGFR) has been observed and the activation of Akt and p38 pathways were involved [13]. Since TBLF and rTBL-1 are molecules with cytotoxic activity and potential for use as chemotherapeutic tools, it is necessary to understand the adjacent mechanisms they evoke in cells—such as the activation of second messengers like cAMP, IP_3_, DAG, or calcium—have not been studied.

Particularly, the calcium ion, besides being a second messenger, is a chemical element with a divalent charge and an important biochemical effector. Its nature makes it one of the most versatile factors in achieving a variety of different cellular events, from transmitter release to metabolic regulation, cell cycle control, cell differentiation, and cellular programming [14,15,16,17,18,19]. A large number of stimuli influence the increase in cytosolic calcium concentration ([Ca^2+^]_i_) and the release of calcium from the endoplasmic reticulum (ER) [14,20,21,22,23,24]. Therefore, cells are constantly working to maintain the correct concentration gradient. Various studies have found that many cytotoxic agents cause uncontrolled calcium entry into the cell, as well as the overstimulation of receptors [19,25,26,27].

Few studies address the effect of lectins in calcium mobilization such as Concanavalin A, which was studied in human platelets where an increase in cytoplasmic calcium concentration was observed as the result of calcium mobilization from internal stores [28]. Calcium mobilization was studied by flow cytometry in a murine macrophage/monocyte cell line, J774A.1, treated with Mistletoe lectin ML-1 [29]. Lectins also stimulate lymphocyte proliferation by the uptake of extracellular calcium [30,31]. However, the effects of lectins on cellular calcium dynamics have been not studied enough. Here, we report the effect of TBLF and rTBL-1 on cellular calcium mobility and its relation to cell viability in cancer and not cancerous breast cells.

## 2. Results

### 2.1. TBLF and rTBL-1 Equivalences

TBLF corresponds to a semi-pure native lectins fraction of Tepary beans, where two main lectins have been identified, TBL-1 and TBL-2 (Figure 1), and some minor high-molecular-weight proteins that, when tested separately, did not show cytotoxic effects on cells [10]. The major band protein, TBL-1, corresponds to 41% of the total proteins and it is homologous to the recombinant lectin, rTBL-1, which is equivalent to 75% of the total protein of TBLF as has also been observed previously [12]. rTBL-1 conserves a 96.24% homology with the TBL-1 peptide sequence, and the differences in molecular weight are due to the addition of a 6XHis tag (~840.92 Da); the sequence EAEAAA at the N-terminal, derived from the Kex2p cleavage of the α-factor (~561 Da); and the presence of glycosidic antennas ~640 Da bigger than those on TBL-1 [11]. Despite the differences, rTBL-1 conserves its cytotoxic effect on cancer cells [12,13].

### 2.2. TBLF Induces an Increase in [Ca^2+^]_i_ and Calcium Release from Intracellular Stores in MCF-12F Cells

Calcium mobilization in MCF-12F breast cells is expressed as arbitrary units of increased fluorescence of the Fluo-4 calcium indicator (AU, Y-axis) versus time (X-axis). Figure 2 shows a representative example of a dose–response curve to TBFL, using three concentrations: 0.12, 1.74, and 27.85 µg/mL.

Figure 3 shows images of the response of MCF-12F cells to TBLF and the analysis of the kinetics of the applied doses. In Figure 3A–C, we show representative images where the cells’ response to the applied dose can be observed.

Figure 3 shows images of the response of MCF-12F cells to TBLF and the analysis of the kinetics of the applied doses. In Figure 3A–C, representative images are shown, where the maximum response of the cells to the applied dose can be observed. In Figure 3A’–C’, the graphs display how the calcium kinetics change, from the basal level to the peak of the response, with the applied TBLF dose. We determined the average activation constant (τ_on_) of the calcium signal induced by different TBLF concentrations using the following equation: y = A1 + (A2 − A1)/(1 + 10^((LOGx0-x)*p)), where LOGx0-x determines the mean activation time of the response (ton) induced by a specific TBLF concentration. We also calculated the Hill constant to assess possible cooperativity in the process. At low doses, the response develops slowly but progressively increases with higher concentrations of TBLF until reaching a peak. In Figure 3A”–C”, the data were normalized by considering the basal average before the TBLF application and the maximum response induced by the lectin addition. It was observed that approximately 90% of the MCF-12F cells responded to TBLF (Figure 3A’’’–C’’’). All these analyses indicate that the increase in cytosolic calcium, directly related to the increase in fluorescence, rose with the TBLF dose and with time. We determined that the EC50 is 4.54 mg/mL of TBLF, while τ_on_ is inversely proportional to the response: higher lectin doses result in a lower τ_on_ (from 1.25 min to 0.6 min), while cooperativity, as indicated by the Hill constant, increases from 0.37 to 1.0. These data indicate that TBLF induces a dose-dependent increase in cytoplasmic calcium concentration, with higher doses generating a faster and more significant increase in [Ca^2+^]_i_.

The increase in fluorescence induced by TBLF does not determine the source of calcium (whether from extracellular entry or release from intracellular stores), so we conducted experiments in low-extracellular-calcium conditions to determine the contribution of the endoplasmic reticulum (ER). Cells were incubated with Fluo-4 AM under normal conditions, but at the start of the experiment, the Krebs solution (KS) was replaced with calcium-free KS. Under these conditions, cells were challenged with 1.74 and 27.85 µg/mL of TBLF. Figure 4A summarizes the response; the left graph shows the typical response for the concentrations used of TBLF in a normal KS medium, while the right graph shows the response in low-extracellular-calcium conditions. Both concentrations induced a sustained fluorescence increase until reaching a peak, which then decreased over time. The response is dose-dependent and biphasic and tends to return to basal levels depending on the dose used. In Figure 4B, we normalized and analyzed experimental data using the same criteria as in the previous and compared the response to TBLF in the absence of extracellular calcium. We found that for MCF-12F cells stimulated with TBLF, the τ_on_ for 27.85 mg/mL was 0.8 min in the presence of extracellular calcium and 2.0 min without extracellular calcium. For 1.74 mg/mL, the τ_on_ was 1.0 min with extracellular calcium and 1.4 min without. These data indicate that TBLF can induce calcium release in a dose-dependent manner, though the mechanism of this event remains unknown.

### 2.3. TBLF-Induced [Ca^2+^]_i_ Mobilization Originates from the Endoplasmic Reticulum Through the IP3 Receptor Pool

Figure 5 shows the increase in intracellular calcium [Ca^2+^]_i_ using the Fluo-4 AM probe in the MCF-12F mammal cell line. To determine whether the Ca^2+^ elevation promoted by lectins involves the ER, we decided to investigate the depletion of ER Ca^2+^ pools associated with IP_3_. The experiment was conducted in Ca^2+^-free KS supplemented with 2 mM EGTA. The stimulation with TBLF at high concentrations (27 µg/mL; arrowhead 1) induced an increase in [Ca^2+^]_i_; however, extrusion mechanisms did not restore normal Ca^2+^ levels. Under this condition, a second application (27 µg/mL) produced a slower response (arrowhead 2), as the pool had been partially depleted of Ca^2+^. Subsequently, stimulation with ATP (200 µM) to activate the formation of the second messenger IP_3_ through the activation of P2Y-type purinergic receptors (PLC-IP_3_) showed that the ER reservoir was completely emptied, as no notable increase in the Fluo-4 AM signal was observed. Under normal conditions, this signal reaches 200 A.U. as shown in Figure 2, whereas in these experiments, it reached approximately 40 A.U. Finally, we added 2 mM extracellular Ca^2+^ (arrowhead 4) to activate store-operated calcium entry (SOCE). SOCE is a cellular response that reflects the [Ca^2+^] ER status. Similar experiments in which we applied ATP twice prior to stimulation with TBLF showed similar results (Material intended for publication). These results suggest that the IP_3_-sensitive Ca^2+^ pool in the ER participates in the elevation of [Ca^2+^]_i_ induced by TBLF.

### 2.4. [Ca^2+^]_i_ Modulation by TBLF and rTBL-1 in MCF-12F and MCF-7 Cells

In Figure 6, we show the response to increasing doses of TBLF (blue lines) and rTBL-1 (red lines) in MCF-12F (column A) and MCF-7 (column B) cells. The top graphs present responses obtained for 10 µg/mL; the central graphs show kinetics for 100 µg/mL; the bottom graphs show the response at 1 mg/mL. For MCF-12F cells, the response was concentration-dependent and biphasic for both types of lectins. However, although both lectins could elevate [Ca^2+^]_i_ in MCF-7 cells, the response kinetics were different (slow and sustained), concentration-dependent, and not biphasic, and the cells did not return to their basal state; these parameters were similar for both lectins in this cell line. These results demonstrate that both lectins can increase [Ca^2+^]_i_, though it is likely that the cellular mechanisms by which they activate this cation are cell-specific.

### 2.5. TBLF Decreases Cell Viability (CVi) of MCF-7 and MCF12F Clones

The Metabolic Cellular Activity (MCA) of MCF7 and MCF12 cells incubated with TBLF was determined using the AlamarBlue™ Cell Viability Reagent method. This method monitors Metabolic Cellular Activity (MCA) by reducing the non-fluorescent indicator resazurin to fluorescent resorufin, where metabolic activity is related to cell viability loss, resulting in no reduction in the indicator and no fluorescence signal generation. A concentration curve using sublethal concentration [9] based on the order of magnitude (untreated, 1, 10, 100 ng/mL; 1, 10 µg/mL) and lethal concentration in the order of magnitude (100 µg/mL and 1 mg/mL) was performed, including positive controls (5 mM glucose) and negative controls (5 µM thapsigargin (TG) and 0.1% Triton X-100 (TX-100)). Fluorescence signal generation was monitored at 10 min and 6, 12, 24, 48, and 72 h.

In Figure 7, the graphs on the left correspond to the non-transformed MCF-12F cells, and those on the right correspond to MCF-7 cancer cells. In the Y-axis, we show the percentage of MCA expressed as a percentage of cellular viability (CVi), and in the X-axis, the different treatments used. Panel 6A shows the graphs for 10 min of incubation; the results indicate that viability was not compromised in both cell lines at any TBFL concentration. Panel 6B corresponds to 6 h of incubation, showing differential effects of treatments between clones. For MCF-12F cells, viability decreased for all the TBLF concentrations tested, while untreated and glucose controls, maintained MCA and negative controls showed a decrease in MCA in the order of TX-100 > TG, which remained throughout the experimental time. The MCA decrease in TBLF-treated cells could be related to a high response to the changes in the intracellular calcium concentration. For MCF-7 cells, there were no significant changes in CVi in the dose–response curve; untreated and glucose controls showed no significant changes, while negative controls showed MCA in the order of TX-100 > TG, as expected. It has been observed that MCF-7 cells are refractory to TBLF treatment because they do not achieve the normal rheostasis of intracellular calcium concentration; therefore, cytoplasmic calcium remains high all the time. In fact, in the case of MCF-7 cells, the response over time did not exhibit important changes.

Panel 7C corresponds to 12 h of incubation and shows that MCF-12F cells exhibited a dose-dependent decrease in cell viability; positive proliferation controls remained at 100%; and TG and TX-100 continued to decrease. For MCF-7 cells, the dose effect was more evident, with MCA decreasing as TBLF concentration increased, while CVi in TG and TX-100 controls decreased. This trend continued up to 24 h in both clones. Finally, at 48 and 72 h, both cell types showed MCA recovery at low concentrations (1–100 ng/mL) and to a lesser extent at medium concentrations (1 and 100 µg/mL). In the case of MCF-12F cells, sublethal concentration showed a recuperation, probably due to the normalization of calcium intracellular concentration, while the higher concentrations remained low. These data indicate that MCA and, consequently, cell viability are affected by TBLF in both cell types, in a differential way that is related to intracellular calcium mobility.

## 3. Discussion

Lectins are mainly glycoproteins that specifically recognize cell membrane glycans [4,32]. As cancer cells exhibit changes in their glycosylation patterns, there is strong evidence that some lectins can affect cancer cell proliferation and viability by apoptosis induction [33,34]. Particularly, TBLF induces apoptosis in colon cancer cells in a differential manner [7], and such an effect was also observed for rTBL-1, where it was possible to determine the interaction with the epidermal growth factor receptor (EGFR) [13], which is one of the main targets on several cancers due to its glycosylation being related to cancer progression [35].

Calcium dynamics depend on the extracellular calcium influx and on the intracellular calcium stores and, therefore, from the activation of membrane receptors such as G-protein-coupled receptors and transient receptor potential channels [36,37]. Particularly, cross-talk between the calcium-sensing receptor and the EGFR in MCF-7 breast cancer cell proliferation has been observed [38], and cooperative signaling between calcium and EGF on tumor cells has also been observed [39].

Our experiments demonstrate that TBLF is capable of inducing changes in [Ca^2+^]_i_ in MCF-12F and MCF-7 cells, being the first study to describe this cellular event triggered by Tepary bean lectins. The responses are concentration-dependent, with a positive correlation in fluorescence increase as an indirect measure of calcium mobilization, in function with TBLF concentration. The increase in [Ca^2+^]_i_ triggered by TBLF suggests that the response is generated by the activation of cellular mechanisms involving calcium ions (ON mechanisms), where calcium channels, non-selective cation channels, exchangers, and the activation of membrane receptors that induce the generation of second messengers culminating in calcium release from intracellular stores (mainly ER) are involved. Since this cation participates in various cellular processes, its persistence in the cytoplasm is highly regulated. Therefore, the calcium signal must be turned off (OFF mechanisms) by the coordinated action of organelles and proteins: calcium reuptake into organelles, extrusion by plasma membrane or mitochondrial pumps and/or exchangers, and sequestering proteins [15]. Thus, in MCF7 and MCF-12F cells, TBLF modifies the homeostasis of this cation in the cytosol, indicated by calcium accumulation in the cytoplasm and, subsequently, the mechanisms that the cells possess to reduce this until the signal is turned off. As observed in our experiments, both cell clones are responsive to TBLF: the response is dose-dependent and promotes calcium release from the ER. However, in MCF-12F cells, the response is biphasic, while in the MCF-7 clone, the response develops gradually, suggesting either that the proteins involved in the ON and OFF mechanisms that each cell line exhibits are different or that the cellular process is affected [19,20,40].

We determined the activation time τ_on_, Hill, and EC_50_ of TBLF in MCF-12F cells, where the signal values were considered: in the basal region and at the peak of fluorescence (as shown in Figure 2, the response peak was reached independently of the TBLF concentration used). The values were adjusted to the Boltzmann equation, indicating that at higher TBLF concentrations, the response is faster. The data were also adjusted to the Hill equation to determine the cooperativity of response activation. The adjustment results revealed that at higher TBLF concentrations, the activation response time is shorter, and positive cooperativity is present in the process. Additionally, the reproducibility of the event is high; we conducted the analysis on 700 cells from seven independent cultures, but approximately 10% of the cells were non-responsive regardless of the treatment, as we had reported previously [19]. This type of analysis was not possible in MCF-7 cells due to the kinetics developed by the TBLF application, supporting the existence of differences in the mechanisms involved in calcium homeostasis in each cell line [19,20].

The application of the Hill equation to our data indicates that cooperativity exists in the increase of Ca^2+^ as a function of TBLF concentration in MCF-12F cells. However, this equation does not allow us to distinguish how much Ca^2+^ enters from the extracellular space and how much Ca^2+^ originates from ER release as shown in Figure 6. To address this question, we conducted experiments under low-extracellular-calcium conditions. We determined calcium mobilization with TBLF (27.85 and 1.75 µg/mL) under two conditions: normal (2 mM CaCl_2_) and zero (0 mM CaCl_2_, 2 mM EGTA). In this way, the recorded Ca^2+^ necessarily comes from internal stores such as the ER.

The results demonstrated that for both TBLF concentrations in the presence of extracellular calcium, the cellular response was as expected. However, under zero-extracellular-calcium conditions, fluorescence intensity decreased but retained its biphasic nature at both concentrations. This observation suggests that TBLF induces calcium release from endoplasmic reticulum (ER) stores, likely mediated by second messengers such as inositol triphosphate (IP_3_). IP_3_ activates its receptor (IP_3_R) on the ER membrane, leading to calcium release, or alternatively, calcium release may occur via ryanodine receptor activation. As expected, calcium extrusion mechanisms (“OFF” mechanisms) were more effective under zero-extracellular-calcium conditions, as the contribution of calcium from the ER is limited. In this scenario, excess calcium is efficiently recovered by ER calcium ATPase (SERCA) into intracellular stores [19,20] or expelled to the mitochondria or extracellular space via extrusion pathways [18].

In summary, while TBLF primarily triggers a massive calcium influx from the extracellular space, ER-stored calcium also contributes to the response. Additionally, the observed increase in τ_on_ values at both concentrations indicates a delay in the response, supporting the hypothesis that TBLF interacts with a receptor-type protein. This delayed response likely results from intracellular signal transduction mechanisms. Further experiments are required to elucidate the specific pathways involved.

Under normal conditions (2 mM extracellular CaCl_2_), the application of increasing TBLF doses (27, 85, and 1.75 µg/mL) elicited the expected responses (Figure 2). In contrast, under zero-extracellular-calcium conditions, several distinct events were observed: (1) The basal fluorescent signal of the calcium reporter decreased significantly. (2) Saturating concentrations of TBLF induced a biphasic response, starting with calcium release from the ER and a subsequent increase in cytosolic calcium concentration ([Ca^2+^]_i_). As expected, calcium extrusion mechanisms were activated to maintain homeostasis by recovering excess calcium into the ER, exporting it to the extracellular space, or sequestering it into mitochondria [18,19,20]. (3) A second application of TBLF resulted in a weaker response, indicating the partial depletion of ER calcium stores, which limited subsequent TBLF-induced responses. (4) To determine if the activated calcium pool is IP_3_-sensitive, we applied a purinergic agonist that stimulates P2Y receptors. This pathway activates phospholipase C (PLC), generating IP_3_, which targets IP_3_ receptors (IP_3_Rs) in the ER membrane [20,21,22]. The lack of response to purinergic stimulation confirmed that TBLF had depleted the IP_3_-sensitive calcium pool. (5) In the final phase, we restored extracellular calcium levels by adding 2 mM CaCl_2_. If cells detect abnormally low ER calcium levels, store-operated calcium entry (SOCE) is activated. The resulting massive calcium influx from the extracellular space increased Fluo-4AM fluorescence, reflecting calcium uptake into the cytosol and its subsequent transfer into the ER by membrane-bound calcium transport proteins. These findings strongly suggest that TBLF activates the IP3-sensitive calcium pool in the ER, as evidenced by the biphasic calcium response and depletion of this intracellular store (Figure 4 and Figure 5).

These experiments open the possibility that, under extracellular calcium conditions, TBLF and rTBLF can induce a significant calcium event that facilitates intra- and intercellular communication, supported by the CICR phenomenon. Our results show that TBLF and rTBL-1 efficiently mobilize calcium in both cell lines, although with significant differences in calcium homeostasis mechanisms between non-cancer cells (MCF-12F) and cancer cells (MCF-7). These differences indicate that cancer cells maintain elevated intracellular calcium levels without regulatory capacity, contributing to their associated cellular effects.

We were able to determine whether TBLF and rTBL-1 can mobilize calcium in both cell lines, and a comparison of the responses in MCF-12F and MCF-7 cells was performed. In MCF-12F cells, the responses triggered by both lectins are similar: biphasic, dose-dependent, and with no significant differences in constants (Hill, EC_50_, and τ_on_). In transformed cancer cells (MCF-7), the responses are sustained and dose-dependent for both lectins. These findings strongly suggest that TBLF and rTBL-1 can provoke changes in [Ca^2+^]_i_ in both clones. However, the recognition of lectins by membrane receptors, intracellular receptors, and/or proteins involved in calcium dynamics in MCF-7 cells is different. Although both clones have the same embryonic origin, the modifications in MCF-7 due to being derived from metastatic cancer have caused genomic and structural changes and possible mutations in proteins responsible for intracellular calcium homeostasis [19]. It is important to notice that MCF-7 cells did not respond equally to TBLF and rTBL-1 since rTBL-1 corresponds only to the majority lectin of TBLF, and it is known that mixtures of bioactive compounds usually enhance biological responses [41].

To determine whether TBLF affects cell viability, we performed experiments to quantify MCA as an indirect measure of cell viability under different treatments. Generally, it was observed that TBLF affects the viability of both cell lines in a differential way. MCF-12F cells showed a high decrease in CVi percentage until 24 h due to the increment of intracellular calcium by the effect of TBLF. However, after 24 and until 72 h, MCA recovered from the TBLF challenge as a function of TBLF concentration. In contrast, MCF-7 cells showed a lower response that remained throughout the experiment, and only the highest concentrations exhibited a decrease in CVi percentage. We suggest that this difference between cell lines is related to the ability of each one to achieve [Ca^2+^]_i_ homeostasis through the remodeling of the battery of proteins involved in this process, provoking calcium extrusion to intracellular reservoirs such as the ER, mitochondria, or extracellular space. Such a mechanism is affected in MCF-7 cells and, therefore, is not able to respond within the first hours to the TBLF treatment.

We suggest that in healthy cells, lectin exposure activates the elevation of intracellular calcium, but the cells have organelles and proteins that can counteract this effect. On the other hand, altered calcium homeostasis mechanisms present in transformed cancer cells are resistant to lectins and highly efficient, activating mechanisms that favor their phenotype. However, it is a fact that molecules such as TBLF and rTBL-1 lectins decrease cell viability. Additionally, conducting transcriptomic studies to investigate possible alterations in gene expression levels at different times (early, intermediate, and late) to establish some molecular markers that either decrease or regulate their expression in response to the challenge would be beneficial.

In conclusion, our results show that both TBLF and rTBL-1 efficiently mobilize calcium in both cell lines, where at varying concentrations, cytosolic calcium increases in both cell lines. In addition, we have been able to determine the activation time of these responses, which differs depending on the concentration used. We have identified extracellular calcium concentration as a key factor influencing cytosolic calcium levels. Furthermore, we show that even when extracellular calcium is removed, cytosolic calcium continues to increase, indicating that a mechanism is activated to release calcium from intracellular stores. Both TBLF and rTBL-1 significantly elevate calcium levels, leading to the generation of intracellular and intercellular calcium waves, triggered by events already described and known as Calcium-Induced Calcium Release (CICR). Interestingly, cancer cells respond differently to lectin (TBLF and rTBL-1) stimuli, showing no calcium rheostasis and maintaining elevated levels of intracellular calcium, with its associated cellular effects. Pharmacological experiments to inhibit endoplasmic reticulum calcium-releasing receptors (IP3R and RyR) as well as their reuptake by SERCA would help us determine more detailed mechanisms. In future research, we will be looking at experiments that point in that direction.

## 4. Materials and Methods

### 4.1. Tepary Bean Lectin Fraction (TBLF) Production

TBLF was obtained from Tepary bean seeds according to a previously reported method [7]. Briefly, the seeds were ground, and the obtained flour was degreased using CHCl_3_/MeOH (3:1), washing it several times until the filtrate was clear. Protein extraction was performed using Tris-HCl pH 8.0 at 4 °C, and precipitation was achieved using ammonium sulfate in two steps, from 40% to 60% (*w*/*v*) saturation. Protein was dialyzed against tridistillated water and followed by molecular weight exclusion chromatography using a Sephadex G-75 column (Thermo Fisher, Waltham, MA, USA). TBLF was lyophilized and stored at −20 °C until use.

### 4.2. rTBL-1 Production

Pichia pastoris SMD1168H yeast was provided by Dr. Elaine Fitches, Department of Biosciences, Durham University, Durham DH1 3LE, UK. The heterologous expression and purification of rTBL-1 were performed following the reported methodology [11,12]. The culture medium was centrifuged (30 min, 7500× *g*, 4 °C), and the supernatant was clarified and filtered using 2.7 and 0.7 μM glass fiber filters (Whatmann, Maidstone, UK). Purification was carried out by the supernatant using HisTrap HP nickel affinity columns (GE Healthcare, Maidstone, UK), followed by dialyzation and lyophilization.

### 4.3. Protein Quantification

Protein was quantified using the Bradford method [42] followed by SDS-PAGE [43]. Normalization of TBLF and rTBL-1 lectin was conducted by protein band densitometry from 12% (*w*/*v*) polyacrylamide SDS-PAGE.

### 4.4. Cell Culture

Cell culture was conducted according to previously reported techniques [19]. Briefly, healthy cultures of MCF-17 (ATCC HTB-22) and MCF-12F (ATCC CRL-3599) were maintained in Dulbecco’s modified Eagle medium (DMEM, 12100-046, Gibco, Waltham, MA, USA) supplemented with 10% fetal bovine serum (FBS, P30-3306, PAN Biotech, Aidenbach, Germany) and 1X antibiotic–antimycotic (Gibco) in a controlled CO_2_ atmosphere at 37 °C. Confluent cultures were treated with Versene (Thermo Fisher, 15040066), followed by 0.1% trypsin, and were subsequently transferred to 96-well plates or sterile round coverslips measuring 1 cm^2^. The cells were maintained in culture for 18–24 h and then used for various assays.

### 4.5. Intracellular Ca^2+^ Mobilization

Changes in [Ca^2+^]_i_ in MCF-12F and MCF-7 cells upon lectin stimulation were recorded in cultures seeded on round coverslips at a density of 0.6–0.8 × 10^6^ cells. Cells were incubated with 5 µM of Fluo4-AM (Thermo Fisher, No. F14201) in Krebs solution (KS, containing in mM, 150 NaCl, 1 KCl, 1 MgCl_2_, 2 CaCl_2_, 4 Glucose, 10 HEPES; pH 7.4, 0.5% bovine serum albumin (BSA), and 0.01% pluronic acid) for 15 min at room temperature. Krebs solution was without calcium (zero [Ca^2+^]) (KS, containing in mM, 150 NaCl, 1 KCl, 1 MgCl_2_, EGTA 2 mM, 4 Glucose, 10 HEPES; pH 7.4, 0.5% bovine serum albumin (BSA)). After incubation, cells were washed three times with KS/0.5% bovine serum albumin (BSA) to remove unincorporated Fluo4-AM and were placed in KS/0.5% BSA for the complete de-esterification of Fluo4-AM. The coverslips were mounted in a homemade recording chamber and observed using an Olympus MVX10 MacroView microscope equipped with an X-Cite 120Q fluorescence lamp, Olympus HQ Filter Set for GFP (HQ filter set for GFP Ex460-480/Em495-540, DM485) (San Jose, CA, USA), and XM10 monochrome camera (CCD, 1376 × 1032 resolution, monochrome b/w and 100 µs to 160 s exposure time) (San Jose, CA, USA).

For our experiments, real-time calcium dynamics videos were obtained with Olympus Cell P software (San Jose, CA, USA), capturing at 1 frame/s, at 250×, and at 20% power of the X-Cite 120Q. Cells were maintained in KS during recording, and stimulation with various lectin concentrations was applied directly. Videos were saved in avi format and later converted to seq format using Image Pro Plus version 6.0 (Media Cybernetics. Rockville Pike, Suite 240, Rockville, MD, USA). Fluorescence values from regions of interest (ROIs) in the selected visual field were obtained using Image J^®^ software v.1.5, National Institutes of Health, Bethesda, Rockville, MD, USA). Data were graphed and analyzed with Origin Pro 2021 (Copyright 2012 Origin Lab Corporation, One Roundhouse Plaza, Northampton, MA, USA). Results were reported as the mean fluorescence arbitrary units (A.U.) of the selected ROIs over time, similar to [19].

Analysis and adjustment of experimental functions: General information of the equation used: y = A1 + (A2 − A1)/(1 + 10^((LOGx0-x)*p). Function Name = Dose Resp (dose–response curve with variable Hill slope given by parameter ‘p’.). Names = A1, A2, LOGx0, p. Meanings = bottom asymptote, top asymptote, center, hill slope. [CONSTRAINTS]: [Parameters Initialization], sort(x_y_curve); //smooth(x_y_curve); A1 = min (y_data); A2 = max(y_data); LOGx0 = xaty50(x_y_curve); double xmin, xmax; x_data.GetMinMax(xmin, xmax); double range = xmax − xmin; if (yatxmax(x_y_curve) − yatxmin(x_y_curve) > 0); p = 5.0/range; else p = −5.0/range. [Derived Parameters] span = abs(A1 − A2); EC20 = 10^(LOGx0 + log (0.25)/p); EC50 = 10^LOGx0; EC80 = 10^(LOGx0 + log(4)/p); EC10 = 10^(LOGx0 − log(9)/p); EC90 = 10^(LOGx0 + log(9)/p).

### 4.6. Cell Viability with AlamarBlue^®^

To determine the viability of MCF-7 and MCF-12F cells challenged with lectins, we used AlamarBlue™ Cell Viability Reagent (A50100, Thermo Fisher Scientific Inc., Waltham, MA, USA). This method detects metabolically active cells, reported as a cell viability and proliferation assay. It quantitatively measures the reducing power of living cells on resazurin (indigo color, non-fluorescent, water-soluble, stable in culture medium, non-toxic, cell-permeable, and electron acceptor) to resorufin (pink and fluorescent) [44]. This oxidation–reduction change was used to measure fluorescence changes (using an excitation between 530 and 560 and an emission at 590 nm) in a VarioSkan (Thermo Fisher Scientific, LUX multimode reader, USA). Confluent cultures of each clone were trypsinized, and cells were counted using a Neubauer chamber (Merck (BRAND^®^), Darmstadt, Germany). Cells were seeded at a density of 0.005 × 10^6^ per well in 96-well plates and maintained for 18–24 h under routine culture conditions and then stimulated with increasing concentrations of TBLF (1, 10 ng/mL, 1,10, 100 µg/mL, and 1 mg/mL). Untreated controls and positive viability controls like glucose (promoter of proliferation in endometrial cancer cells) [45] and negative controls such as thapsigargin (inhibits viability of adrenocortical carcinoma cells by inducing apoptosis through JNK signaling) [46] and TX-100 (induces necrotic death in prostate and colon cancer cell lines) [47] were included. Cells were maintained under standard culture conditions throughout the experiment. Oxidation–reduction values were obtained using the VarioSkan reader at 10 min and 6, 12, 24, 48, and 72 h, taking advantage of the low toxicity of AlamarBlue™ Cell Viability Reagent. Finally, results were graphed and analyzed using OriginPro 2021 software (Copyright 2012 OriginLab Corporation, One Roundhouse Plaza, Northampton, MA, USA).

### 4.7. Statistical Analysis

All data were processed using OriginLab 2021. Parametric statistical analysis with standard deviation and Tukey’s post hoc tests were performed, along with mathematical analyses using the Boltzmann equation (dose–response) and Hill equation (cooperativity).

## 5. Conclusions

Lectins are promising biomolecules with potential therapeutic applications for treating transformed cancer cells. Our findings show that TBLF and rTBL-1 mobilize cytosolic calcium concentrations ([Ca^2+^]_i_) in breast cell lines MCF-12F (non-cancerous) and MCF-7 (cancerous) in a concentration-dependent manner, with distinct activation times confirmed via cooperativity analysis. Calcium-free conditions (2 mM EGTA) suggest that [Ca^2+^]_i_ increases are mediated by plasma membrane proteins, initiating signal transduction.

Comparative analysis revealed synchronized, dose-dependent calcium mobilization in MCF-12F cells, while MCF-7 cells required higher lectin concentrations. Both lectins release calcium from the endoplasmic reticulum, underscoring calcium’s role in early responses. Distinct kinetics and cell viability assays highlighted that MCF-12F cells restore homeostasis efficiently, whereas MCF-7 cells show reduced viability only at high lectin doses. These results emphasize the biomedical potential of TBLF and rTBL-1, encouraging further studies on molecular and transcriptional responses in cancer therapy.

## Figures and Tables

**Figure 1 ijms-26-01064-f001:**
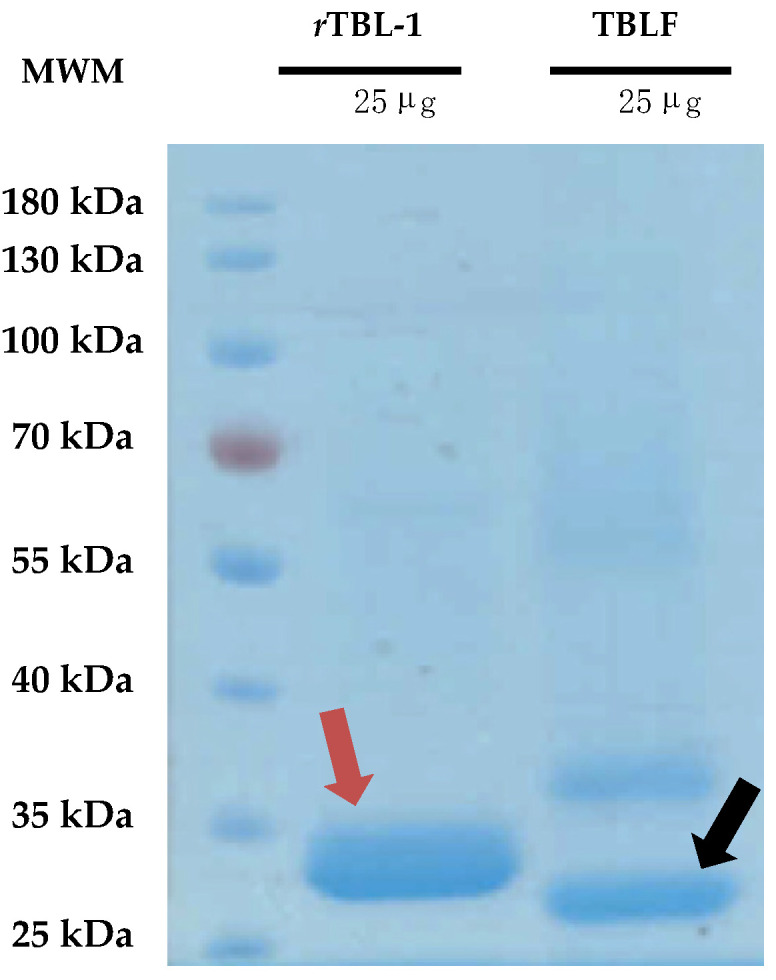
Comparison between the native TBLF and the recombinant lectin rTBL-1. A 12% polyacrylamide SDS-PAGE was performed charging 25 μg of protein of both rTBL-1 and TBLF. Gels were visualized using Coomassie staining. Molecular weight marker, MWM; recombinant lectin, rTBL-1; Tepary bean lectin fraction, TBLF. The red arrow points to rTBL-1 and the black arrow points to the native TBL-1 protein.

**Figure 2 ijms-26-01064-f002:**
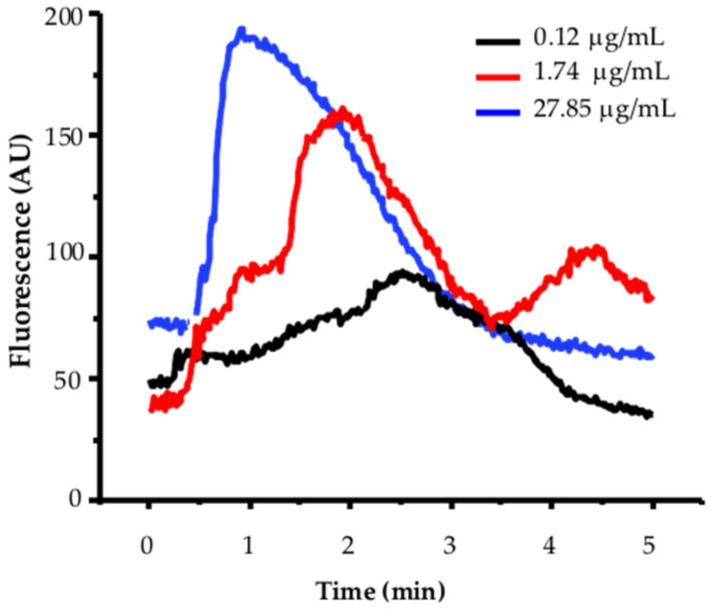
Effect of three different TBLF concentrations of increased cytoplasmic calcium in MCF-12-F breast cells. Cells (600 to 700 from 7 independent cultures) were incubated with 5 mM of Fluo4-AM as outlined in the Section 4 and analyzed on an Olympus MVX-10 fluorescence microscope. In all experiments, activity was recorded for 1 min. Then, increasing concentrations of TBFL were added. Activation induced by a concentration of 0.12, 1.74, and 27.85 μg/mL are shown. A representative experiment of *n* = 15 of 7 independent cultures is shown.

**Figure 3 ijms-26-01064-f003:**
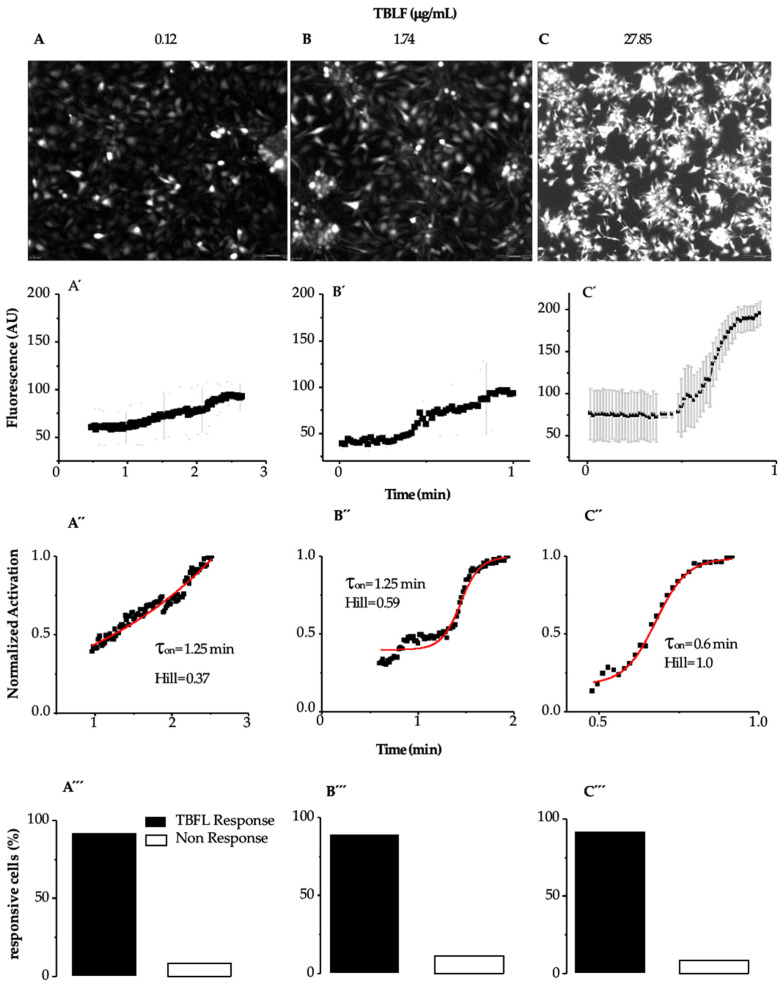
Determination of activation time (τ_on_) induced by TBFL in MCF12-F cells. Experiments were performed using the Fluo 4AM probe as a fluorescent indicator to determine calcium transient events. Representative experiments of *n* = 15 from 7 different cultures are shown. (**A**–**C**) show representative photographs of the maximum fluorescence of each TBLF concentration. (**A’**–**C’**) show the onset of the response to TBFL represented as changes in fluorescence in arbitrary units (AU) as a function of time (averages ± SD are shown). (**A”**–**C”**) show the percentage of cells that responded during the TBFL treatments and those that did not. (**A’’’**–**C’’’**) show that approximately 90% of the MCF-12F cells responded to TBLF.

**Figure 4 ijms-26-01064-f004:**
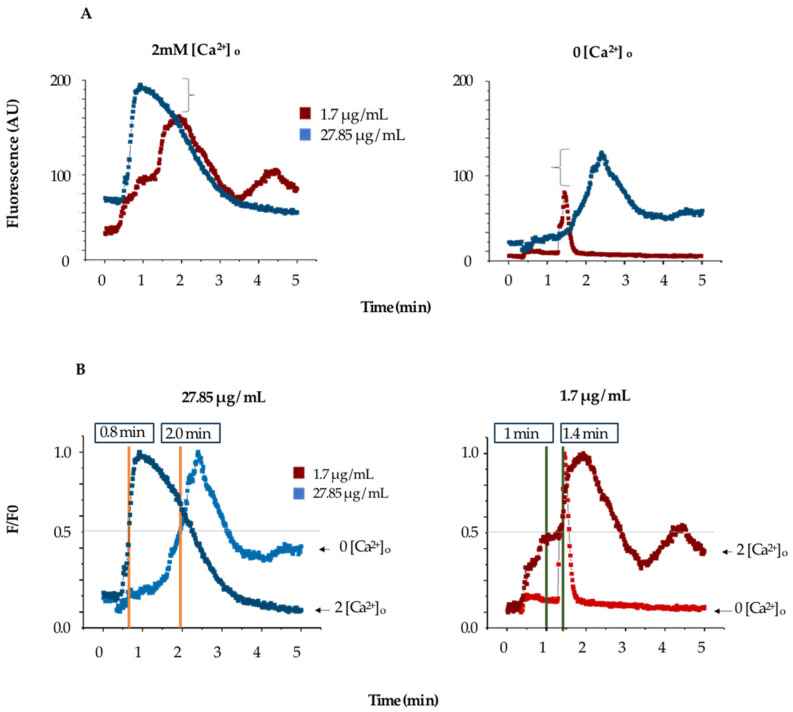
Determination of the contribution of extracellular calcium in the activation of the dose-dependent response to TBFL. (**A**) shows the change in fluorescence induced by two concentrations of TBFL (1.7 and 27.85 μg/mL; red and blue respectively) in the presence of extracellular calcium (left) and without extracellular calcium (right) with 2mM EGTA added. The brackets demonstrate the fluorescence change in both conditions. (**B**) shows the normalized data and comparison of the activation times in the presence and absence of calcium.

**Figure 5 ijms-26-01064-f005:**
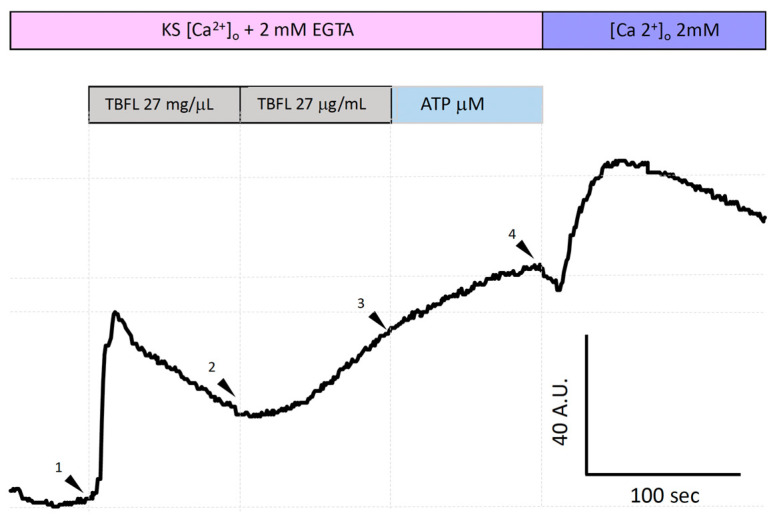
TBLF-induced intracellular Ca^2+^ mobilization from the endoplasmic reticulum in MCF-12F cells. This shows a representative trace of calcium mobilization from the ER in Ca^2+^-free solution with the addition of 2 mM EGTA. Two consecutive applications of 27 µg/mL (arrowheads 1 and 2) were performed, followed by the application of 200 µM ATP (arrowhead 3). Finally, 2 mM extracellular Ca^2+^ was added (arrowhead 4). This is a representative experiment out of a total of 600 cells analyzed from 5 independent experiments.

**Figure 6 ijms-26-01064-f006:**
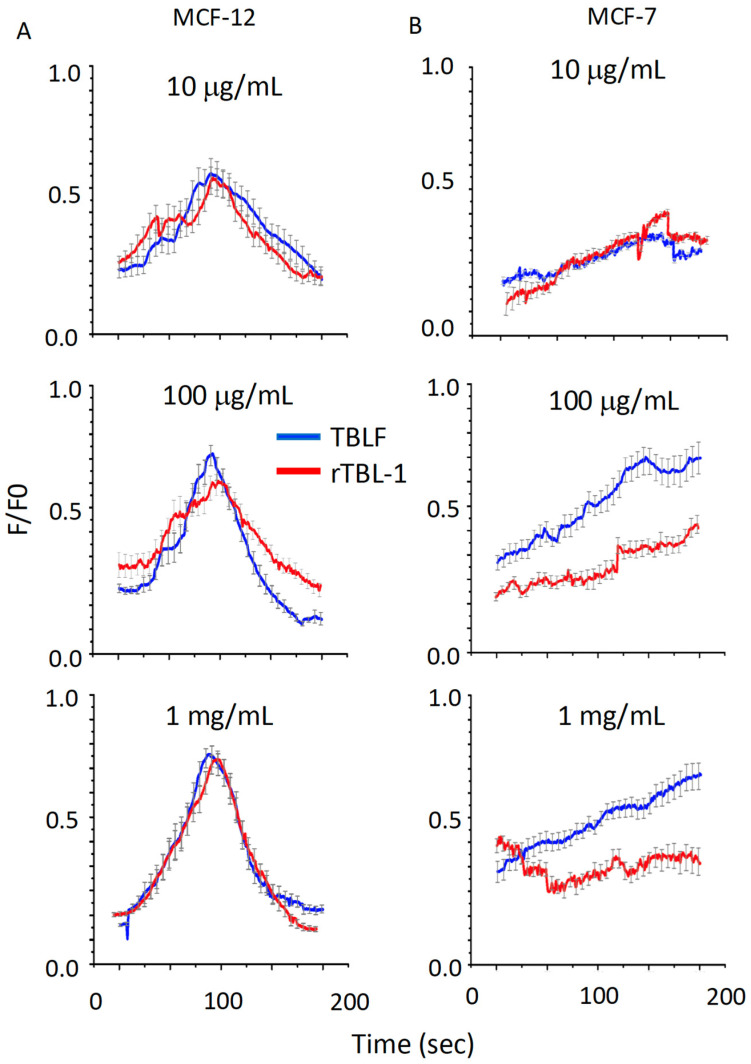
Resolution of calcium signaling activation induced by TBFL and rTBL-1 in MCF-12F breast cell lines and MCF-7 cancer cells. Concentrations of 10, 100 μg/mL, and 1 mg/mL for TBLF in blue and rTBL-1 in red. Column (**A**) shows the results for the MCF-12F line and in column (**B**), those for the MCF-7 line. The change in fluorescence as a function of time is reported. Cells were loaded with Fluo4-AM and the average of 30 cells from 5 different cultures is shown (±standard deviation).

**Figure 7 ijms-26-01064-f007:**
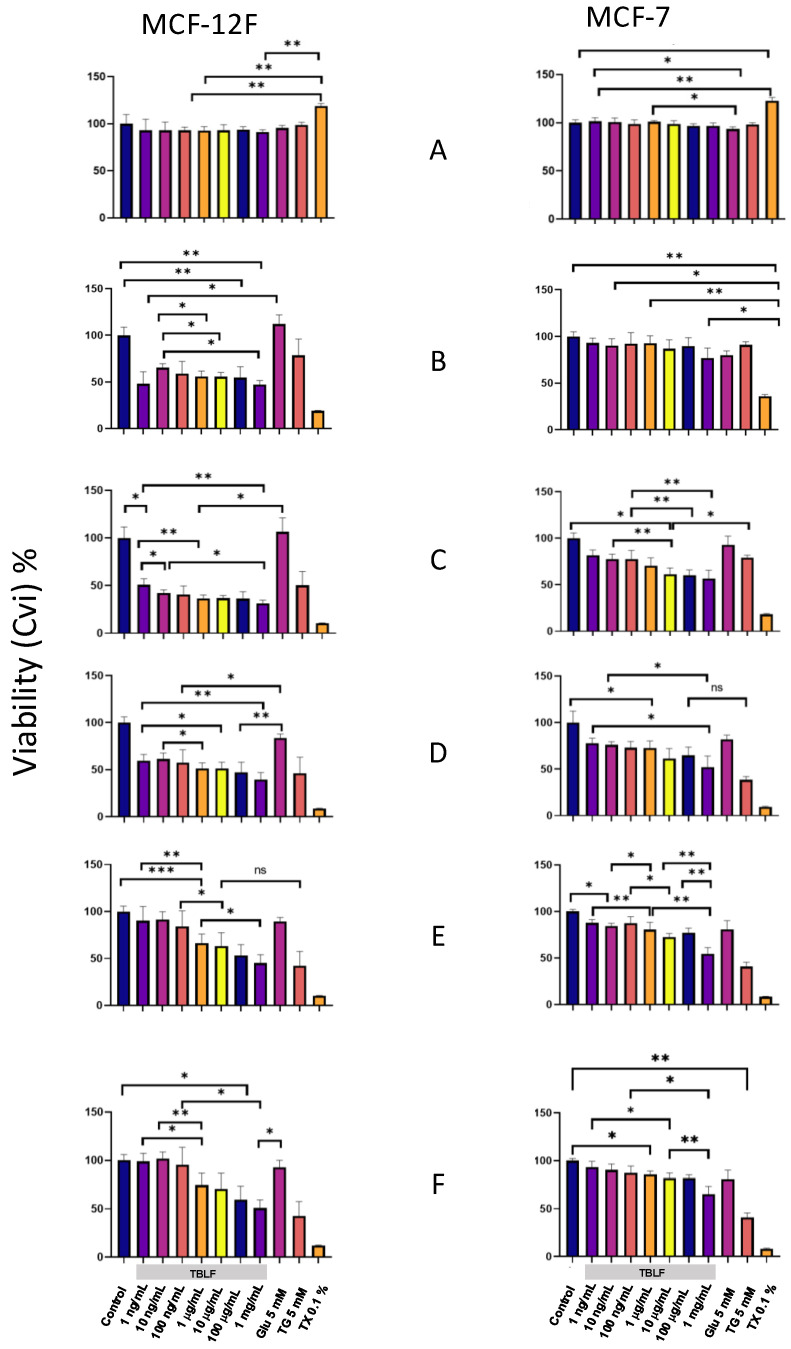
Cell viability in MCF-12F and MCF-7 cells by challenge with increasing concentrations of TBFL. Using the Alamar Blue kit, we were able to measure the conversion of resazurin to resorufin as a direct quantitative measure of cell viability (CVi). Results are shown at 10 min and 6, 12, 24, 48, and 72 h (panels **A**, **B**, **C**, **D**, **E**, and **F**, respectively). The following were used as controls: negative MCF-12F and MCF-7 cells without treatment; positive control 5 mM glucose as energy source to induce cell proliferation; thapsigargin (TG) 5 mM as a control for increased intracellular calcium ([Ca^2+^]_i_), TritonX-100 (0.1%); and increasing concentrations of TBFL: 1, 10, and 10 ng/mL; 1, 10, and 100 μg/mL; and 1 mg/mL. Fluorescence change determinations were performed in a Variosckan spectrophotometer at an excitation wavelength of 560 nm, and emission was collected at 580 nm. The results shown are presented as the means ± S.D., from *n* = 3 independent cultures. The asterisk shows that the mean difference is significant at the * *p* < 0.05, ** *p* < 0.01, *** *p* < 0.001 level from data obtained from Tukey’s post hoc test.

## Data Availability

The data presented in this study are available on request from the corresponding author.

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
