# Peer review of "Tepary Bean (Phaseolus acutifolius) Lectins as Modulators of Intracellular Calcium Mobilization in Breast Cancer and Normal Breast Cells"

_ijms, 2025, doi:10.3390/ijms26031064_

Round 1
Reviewer 1 Report (New Reviewer)
Comments and Suggestions for Authors
This paper investigates the effects of Tepary bean lectins (TBLF and recombinant rTBL-1) on calcium mobilization in breast cancer (MCF-7) and normal breast (MCF-12F) cells. The paper presents compelling evidence that both lectins increase intracellular calcium concentrations dose-dependently, with differential impacts on cancerous and non-cancerous cells. Overall, the manuscript is well designed and organized. Minor comments:
1. Why was MCF-7 chosen as the cancer model, and how do you anticipate other breast cancer subtypes might respond to TBLF?
2. How stable are TBLF and rTBL-1 under physiological conditions (e.g., pH, temperature)?
3. Have you identified specific ion channels or transporters (e.g., TRP channels) involved in the observed calcium responses?
4. Have you explored whether TBLF or rTBL-1 localizes to specific intracellular compartments, such as lysosomes or the nucleus, which might explain their differential effects?
Author Response
Comments and Suggestions for Authors
This paper investigates the effects of Tepary bean lectins (TBLF and recombinant rTBL-1) on calcium mobilization in breast cancer (MCF-7) and normal breast (MCF-12F) cells. The paper presents compelling evidence that both lectins increase intracellular calcium concentrations dose-dependently, with differential impacts on cancerous and non-cancerous cells. Overall, the manuscript is well-designed and organized. Minor comments:
- Why was MCF-7 chosen as the cancer model, and how do you anticipate other breast cancer subtypes might respond to TBLF?
In our research group, we have characterized calcium mobilization in MCF-12 and MCF-7 cells. Over several years, we have studied calcium mobilization from intracellular reservoirs, such as the ryanodine receptor and the IP3 receptor (Saldaña C, 2009). Based on these studies, and in agreement with other research groups, we have proposed that Ca2+ signaling is essential for cell proliferation and appears to play a key role in cancer development, tumor invasiveness, and drug resistance (Butanda-Ochoa 2006; Kunzelmann 2005; Lang F 1991).
On the other hand, MCF-7 breast cancer cells have been shown to be sensitive to the cytotoxic effect of TBLF, as shown by García-Gasca et al. (2012) where the effect was concentration dependent with a LC50 of 4.99 AU/mL, being one of the most sensitive cell lines after ZR-75-1 breast cancer cells with an LC50 of 4.71 and CaCo2 colon cancer cells with a LC50 of 0.15 AU/mL. Research with breast cancer cells was not continued because the strategy against colon cancer in vitro and in vivo by intragastric route was developed. A text specification was added in lines 55-56.
Butanda-Ochoa, Armando & Höjer, Germund & Morales-Tlalpan, Verónica & Díaz-Muñoz, Mauricio. (2006). Recognition and Activation of Ryanodine Receptors by Purines. Current medicinal chemistry. 13. 647-57. 10.2174/092986706776055715.
García-Gasca, T.; García-Cruz, M.; Hernandez-Rivera, E.; López-Matínez, J.; Castaneda-Cuevas, A.L.; Yllescas-Gasca, L.; Rodríguez-Méndez, A.J.; Mendiola-Olaya, E.; Castro-Guillén, J.L.; Blanco-Labra, A. Effects of Tepary bean (Phaseolus acutifolius) protease inhibitor and semipure lectin fractions on cancer cells. Nutr. Cancer 2012, 64, 1269–1278. DOI:10.1080/01635581.2012.722246
Kunzelmann, Karl. (2005). Ion Channels and Cancer. The Journal of membrane biology. 205. 159-73. 10.1007/s00232-005-0781-4.
Lang, F & Friedrich, F & Kahn, E & Wöll, Ewald & Hammerer, M & Waldegger, S & Maly, Konstantin & Grunicke, Hans. (1991). Bradykinin-induced Oscilations of Cell Membrane Potential in Cells Expressing the Ha-ras-Oncogene. The Journal of biological chemistry. 266. 4938-42. 10.1016/S0021-9258(19)67739-2.
Saldaña, Carlos & Díaz-Muñoz, Mauricio & Antaramian, Anaid & Gallardo, Adriana & Garcia-Solis, Pablo & Morales-Tlalpan, Verónica. (2009). MCF-7 breast carcinoma cells express ryanodine receptor type 1: Functional characterization and subcellular localization. Molecular and Cellular Biochemistry - MOL CELL BIOCHEM. 323. 39-47. 10.1007/s11010-008-9962-7.
- How stable are TBLF and rTBL-1 under physiological conditions (e.g., pH, temperature)?
Previous studies have shown that TBLF is resistant to digestion for up to 72 hours and is capable of activating the immune system (Ferriz-Martínez et al., 2015). In more recent studies, Vega-Rojas et al. (2021) showed that rTBL-1 is able to resist passage through physiological conditions in an in vitro digestion study and ex vivo exposure in inverted intestine and reach the small intestine with biological activity, where its bioavailability and permeability were determined. The results showed that rTBL-1 is resistant to gastrointestinal digestion, is bioavailable at the intestinal level and has mainly partial permeability through enterocytes.
Ferriz-Martínez, R.; García-García, K.; Torres-Arteaga, I.; Rodriguez-Mendez, A.J.; Guerrero-Carrillo, M. de J.; Moreno-Celis, U.; Ángeles-Zaragoza, M.V.; Blanco-Labra, A.; Gallegos-Corona, M.A.; Robles-Álvarez, J.P.; et al. Tolerability Assessment of a Lectin Fraction from Tepary Bean Seeds (Phaseolus acutifolius) Orally Administered to Rats. Toxicol Rep 2015, 2, 63–69, doi:10.1016/j.toxrep.2014.10.015.
Vega-Rojas, L.J.; Luzardo-Ocampo, I.; Mosqueda, J.; Palmerín-Carreño, D.M.; Escobedo-Reyes, A.; Blanco-Labra, A.; Escobar-García, K.; García-Gasca, T. Bioaccessibility and in Vitro Intestinal Permeability of a Recombinant Lectin from Tepary Bean (Phaseolus Acutifolius) Using the Everted Intestine Assay. Int J Mol Sci 2021, 22, 1–22, doi:10.3390/ijms22031049.
- Have you identified specific ion channels or transporters (e.g., TRP channels) involved in the observed calcium responses?
Regarding TRP channels (PubMed), only 8 studies related to these channels and MCF-7 cells were reported between 2008 and 2020. These studies mainly highlight the following aspects: (1) their overexpression in breast cancer cells (2008); (2) their role in the over proliferation of breast cancer cells (2009); (3) the detection of high expression levels in tumors, suggesting potential applications in diagnostic and therapeutic approaches (2009, 2011); and (4) the inhibition of TRP channels, which blocks the S phase of the cell cycle and demonstrates their potential role in regulating cellular proliferation. However, to date, there has been no mention of the activation of TRP channels by bioactive compounds. In our research group, we are conducting transcriptomic studies and developing panels of specific responses, such as those that could activate TRP channel-mediated pathways in the presence of bioactive compounds.
- Have you explored whether TBLF or rTBL-1 localizes to specific intracellular compartments, such as lysosomes or the nucleus, which might explain their differential effects?
We have no evidence that lectins can cross cell membranes and localize within intracellular structures. What we propose is that intracellular calcium concentration ([Ca2+]i) fluctuates within a low nanomolar (nM) range in most cells, in contrast to the higher millimolar (mM) levels present in the extracellular space and internal reservoirs. The electrochemical gradient of Ca2+ within the cell must be finely regulated, as an uncontrolled increase in [Ca2+]i is associated with necrotic or apoptotic events. At the same time, this gradient must be carefully adjusted to allow Ca2+ to function as an intracellular messenger. The mobilization of Ca2+ within intracellular compartments is achieved through the coordinated action of ion channels, exchangers, and metabolic pumps. Among the most relevant Ca2+ release channels present in microsomal membranes are the Ryanodine Receptor (RyR) and the IP3 receptor (IP3R).
Our in vitro studies have shown that TBLF affects signaling pathways related to pAkt/Akt and p53 (Moreno-Celis et al., 2017; 2020). Recently, it was shown that rTBL-1 is able of interacting with the epidermal growth factor receptor, affecting signaling pathways via Akt and p38, as well as apoptosis effectors such as caspase 3 and cleaved PARP-1. In this sense, EGFR degradation is induced via a lysosomal pathway (Dena-Beltrán et al., 2023). In the study by Martínez-Alarcón et al., (2024), it was shown that rTBL-1 is able of differentially binding to EGFR glycans depending on the modification of its CBP residues. These results show that the effects of both, TBLF or rTBL-1, are carried out through interaction with membrane receptors such as EGFR and others not yet explored, rather than by their direct effect on intracellular compartments.
Dena-Beltrán, J.L.; Nava-Domínguez, P.; Palmerín-Carreño, D.; Martínez-Alarcón, D.; Moreno-Celis, U.; Valle-Pacheco, M.; Castro-Guillén, J.L.; Blanco-Labra, A.; García-Gasca, T. EGFR and P38MAPK Contribute to the Apoptotic Effect of the Recombinant Lectin from Tepary Bean (Phaseolus Acutifolius) in Colon Cancer Cells. Pharmaceuticals 2023, 16, 290, doi:10.3390/ph16020290.
Martínez-Alarcón, D.; Varrot, A.; Fitches, E.; Gatehouse, J.A.; Cao, M.; Pyati, P.; Blanco-Labra, A.; Garcia-Gasca, T. Recombinant Lectin from Tepary Bean (Phaseolus·acutifolius) with Specific Recognition for Cancer-Associated Glycans: Production, Structural Characterization, and Target Identification. Biomolecules 2020, 10, 1–16, doi:10.3390/biom10040654.
Moreno-Celis, U.; López-Martínez, J.; Blanco-Labra, A.; Cervantes-Jiménez, R.; Estrada-Martínez, L.E.; García-Pascalin, A.E.; De Jesús Guerrero-Carrillo, M.; Rodríguez-Méndez, A.J.; Mejía, C.; Ferríz-Martínez, R.A.; et al. Phaseolus Acutifolius Lectin Fractions Exhibit Apoptotic Effects on Colon Cancer: Preclinical Studies Using Dimethilhydrazine or Azoxi-Methane as Cancer Induction Agents. Molecules 2017, 22, doi:10.3390/molecules22101670.
Moreno-Celis, U.; López-Martínez, F.J.; Cervantes-Jiménez, R.; Ferríz-Martínez, R.A.; Blanco-Labra, A.; García-Gasca, T. Tepary Bean (Phaseolus Acutifolius) Lectins Induce Apoptosis and Cell Arrest in G0/G1 by P53(Ser46) Phosphorylation in Colon Cancer Cells. Molecules 2020, 25, doi:10.3390/molecules25051021.
Reviewer 2 Report (New Reviewer)
Comments and Suggestions for Authors
The manuscript presents interesting results, however I consider that some points for improvement should be reviewed. Thank you.
1) Figure 3 B" and C" show two kinetics, which should be considered. One of them could be calcium entry, which is important to trigger calcium release by calcium. Reducing extracellular calcium is not always the best option to avoid extracellular calcium influx, so I suggest using specific calcium channel blockers.
2) Please review the legend of Figure 3. They do not correspond to each of the sections referred to.
3) Please review Figure 3 for the percentage of responding cells because it does not "visually" match the images presented.
4) Although TBLF and rTBL-1 are mentioned to be similar, Figure 1 and Figure 6 with MCF-7 indicate otherwise. Please consider the structural and functional differences to avoid referring to them as the same.
5) Please review the concentrations indicated in Figure 7. I believe that 1, 10 and 100 should indicate micrograms.
6) I suggest reducing the Conclusions section to a maximum of two paragraphs.
Author Response
Comments and Suggestions for Authors
The manuscript presents interesting results; however, I consider that some points for improvement should be reviewed. Thank you.
- Figure 3 B" and C" show two kinetics, which should be considered. One of them could be calcium entry, which is important to trigger calcium release by calcium. Reducing extracellular calcium is not always the best option to avoid extracellular calcium influx, so I suggest using specific calcium channel blockers.
Being aware that the experiment is conducted in the presence of extracellular calcium, we performed the experiments presented in Figures 4 and 5, in which calcium was excluded from the extracellular solution and a calcium chelator, such as EGTA, was added at a concentration of 2 mM. In the current state of knowledge regarding breast cells, it is not possible to achieve a highly specific blockade of calcium channels due to the extensive diversity of channels present, making it necessary to resort to a generic blocker. In order to address specific mechanisms, as you kindly suggest, we are currently analyzing transcriptomic panels for these cell lines in comparison to the numerous genes that encode different ion channels that could mobilize calcium. For this reason, we decided to use EGTA at this stage, with the goal of determining that the increase in cytosolic calcium primarily originates from intracellular reservoirs whose activation depends on signal transduction.
- Please review the legend of Figure 3. They do not correspond to each of the sections referred to.
The document has been modified as regards Figure 3 and its concordance with the text in lines 132-152.
- Please review Figure 3 for the percentage of responding cells because it does not "visually" match the images presented.
Figure 3 presents, in panels A, B, and C, representative images of the peak response. What we have observed is that some cells, from the onset of the experiment, already exhibit a higher basal calcium level. As fluorescence increases upon reaching the peak of the response, these cells show a greater amount of relative fluorescence. On the other hand, when we mention that approximately 90% of the cells respond to TBLF and 10% do not, we refer to the fact that this 10% of cells do not experience an increase in fluorescence during the experiment in which TBLF is applied. Furthermore, I would like to emphasize that we have analyzed a total of n = 15, corresponding to 7 independent cultures, which implies that an average of 600 to 700 cells was analyzed. From this sample, we conclude that at least 10% of the cells typically do not respond to the treatment. It is important to note that these results are consistent with previous observations in our studies.
- Although TBLF and rTBL-1 are mentioned to be similar, Figure 1 and Figure 6 with MCF-7 indicate otherwise. Please consider the structural and functional differences to avoid referring to them as the same.
The recombinant lectin rTBL-1 corresponds to the main cytotoxic lectin of TBLF. The difference in molecular weights (approx 2.5 kDa) is attributed to three factors: the addition of a 6XHis tag (~840.92 Da), the sequence EAEAAA at the N-terminal, derived from Kex2p cleavage of the a-factor (~561 Da); and the presence of glycosidic antennas ~640 Da bigger than those on TBL-1 (Martínez-Alarcón et al., 2020;). The homology degree of the mature polypeptide sequences between TBLF and rTBL-1 is 96.24% with only five amino acid mismatches: in position 116, R changed to K; in position 157–158, GQ changed to VN; in position 205, R changed to S; and in position 215, T changed to S. Most of the mismatches are located between loops that serve as blade connectors. None of them are located in the proximity of the glycosylation site, nor do they correspond to any of the amino acids involved in the coordination bonding with the ions. Furthermore, none of them correspond to the residues predicted to be involved in carbohydrate binding. It is not expected that these slight variations influence the folding or activity of the protein (Palmerín-Carreño et al., 2021). Regarding its bioactivity, rTBL-1 retains the biological effect of TBLF since its cytotoxic effect is similar (Dena-Beltrán et al., 2023). An explanation was included in lines 94-99.
The differences in the responses in Figure 6 can be attributed to the presence of another lectin in TBLF, since rTBL-1 corresponds only to the majority lectin, and it is known that mixtures of bioactive compounds usually enhance biological responses. However, it is important to highlight that the response on MCF-7 cells is diametrically different from that of their non-cancerous counterpart MCF-12F with both treatments and that both TBLF and rTBL-1 show similar responses. An explanation was included in lines 391-394.
Dena-Beltrán, J.L.; Nava-Domínguez, P.; Palmerín-Carreño, D.; Martínez-Alarcón, D.; Moreno-Celis, U.; Valle-Pacheco, M.; Castro-Guillén, J.L.; Blanco-Labra, A.; García-Gasca, T. EGFR and P38MAPK Contribute to the Apoptotic Effect of the Recombinant Lectin from Tepary Bean (Phaseolus Acutifolius) in Colon Cancer Cells. Pharmaceuticals 2023, 16, 290, doi:10.3390/ph16020290.
Martínez-Alarcón, D.; Varrot, A.; Fitches, E.; Gatehouse, J.A.; Cao, M.; Pyati, P.; Blanco-Labra, A.; Garcia-Gasca, T. Recombinant Lectin from Tepary Bean (Phaseolus·acutifolius) with Specific Recognition for Cancer-Associated Glycans: Production, Structural Characterization, and Target Identification. Biomolecules 2020, 10, 1–16, doi:10.3390/biom10040654.
Palmerín-Carreño, D.; Martínez-Alarcón, D.; Dena-Beltrán, J.L.; Vega-Rojas, L.J.; Blanco-Labra, A.; Escobedo-Reyes, A.; García-Gasca, T. Optimization of a Recombinant Lectin Production in Pichia pastoris Using Crude Glycerol in a Fed-Batch System. Processes 2021, 9, 876. doi.org/10.3390/pr9050876
- Please review the concentrations indicated in Figure 7. I believe that 1, 10, and 100 should indicate micrograms.
Changes have been made to the description of the figure as well as to the figure itself.
- I suggest reducing the Conclusions section to a maximum of two paragraphs.
Thank you for your observation, we have modified the conclusions.
This manuscript is a resubmission of an earlier submission. The following is a list of the peer review reports and author responses from that submission.
Round 1
Reviewer 1 Report
Comments and Suggestions for Authors
Please see attacched file

Author Response
- Section 4.1. Could authors include a supplementary figure of a SDS-PAGE showing the composition/purity of the purified lectins (TBLF and rTBL-1)? Although authors refer to previous publications in which the procedures were described, it is important to know the composition ofthese fractions to be aware of the presence of possible contaminants that could modulate As described in reference 8, the TBFL contains a number of non-lectin proteins. How to discard thatthese non-lectin proteins have an inducing activity for Ca release?
- See figure 1 in the new version of the paper (page 3).
- The Ca conditions handled in the work are high extracellular Ca (equivalent to 2 mM) and 0extracellular Ca (i.e. no Ca is added). However, if you really want to study any process in the absenceof Ca, it is important to include a control with the presence of some chelating agent, such as EDTA orEGTA, since, in the 0 Ca condition, there is always the risk of the presence of contaminating Ca. Therefore, it would be convenient to carry out additional experiments in the presence of chelatingagent to observe if there is still a response to TBLF. This because a large number of stimuli canincrease the cytosolic calcium concentration and the release of calcium from the endoplasmicreticulum (ER).
R: We did not mention this in the original manuscript and overlooked this detail. In the zero calcium experiments, EGTA was added to a final concentration of 2 mM. (See figure 4, page 6)
- The work was performed using TBLF purified from a bean extract, as well as rTBL-1. In both cases the lectins were used at three different doses, analyzing their effect comparatively between two celllines, the non- cancerous MCF-12F and the cancerous MCF-7. In the experiments described in Figures 1- 4 each dose is analyzed as a function of time, however, no dose-response experimentwas performed that at least included 4-6 different concentrations in order to determine theparameters of the kinetics. Despite this, in the text of the manuscript this term "dose- response"appears 5 times: lines 74, 77, 95, 371, and 376.
R: We acknowledge that there was a mistake in the description. The experiments were conducted using different concentrations, and we should not have described our results as involving changes of 6 to 7-fold magnitude. The explanation provided was based on the fact that the production costs of TBLF and rTBL-1 are high. Recently, we received approval for a new grant, which will allow us to invest more resources. We are currently working on a new manuscript that includes more detailed experiments with varying concentrations and accurate dose-response relationships. We have made changes to the above lines and related figures (figure 2, 3, 4 and 5) in the manuscript where we mentioned dose-response and replaced it with increasing concentrations.
- Could the authors include the bibliographic reference(s) from which the equation included in the caption of Figure 2 was taken? Where they mention that several parameters were determined, precisely using the dose-response equation. Nowhere in the text could I find the definition of A1, A2,x, and
R: We explain that this is a fit using the ORIGENLAB programmed and fit the function to a dose-response model. In this new version of the document, we explain each of the elements that make up the equation (see figure 3).
- It is not clear how the graphs 2A'''', B'''', and C'''' were prepared. Do they reflect the effect seen in images 2A', B'', and C''? It was not clear to me how the % of responsive cells is almost identical with the three different doses (0.12, 1.74, and 27.85 μg/mL)? In the discussion section authors mention that: … we conducted the analysis on 700 cells from seven independent cultures, but approximately10% of the cells were non- responsive regardless of the treatment…. I suggest to describe thisprocedure in the methods section.
R: We would like to clarify that there is always a percentage of cells that do not respond to treatment. To maintain transparency in our experiments, we have included this percentage in our data, as it remains constant in all the cultures we perform. Attached is a video of calcium movement monitored in real time by fluorescence microscopy in which it can be seen that there is a population that does not respond to treatment. We refer to this as the non-responsive population.
https://drive.google.com/drive/folders/1wQl7ZI2kEjTq42yMhF2PGs4tyrd3K-IV?usp=drive_link
- Might it be possible to determine which receptor is responsible for the calcium release induced byTBFL, e., IP3R or Ryanodine receptor through the use of specific inhibitors?
R: The effect we observe when applying TBLF and rLTB-1 to the cell lines is substantial. That is, calcium increases efficiently and rapidly floods the cytosol of a cell, generating an intercellular calcium wave. Based on this, it could even be discussed that this is an intracellular calcium wave activated by a calcium-induced calcium release event. As we mentioned previously, we are studying these events using specific drugs for the RyR and IP3 receptors, such as high concentrations of ryanodine and xestospongin to inhibit both receptors, respectively.
Minor details: attended and yellow-marked
- Title: as a suggestion, substitute mobilizing by mobilize
- Lines 78 and 81 respectively in figure legend 1: the concentrations are presented as mM ormg/mL; they must be μM and μg/mL. (now Line 102)
- Line 84, should be 2A, B, and C (Now line 110)
- Line 147, should be ….A shows the change (Now line 180)
- Line 158, should be 10, 100 ng/ml… (Now line 192)
- Line 185, in figure 4 legend, change mg to μg (Now line 229)
- Line 307, should be …followed by SDS-PAGE, eliminate “and” (Now line 367)
- Line 343, should be ….Results were reported as… (Now line 405)
Figures:
In Figure 3B, the presence of the concentrations shown by red and blue colors could be eliminated because the corresponding concentrations for each graph appears above the graph
R has already been corrected
In Figure 5 legend, and in the text (lines 157, 158), authors mention that they used different concentrationsof TBFL, expressed as ng, μg or mg/mL. However, in the X axis of the graphs, they are presented as nM,μM and mM, please select one form or the other to express the concentration of TBFL added because theyare not equivalent.
R: has already been corrected
Thank you very much for your observations.
Reviewer 2 Report
Comments and Suggestions for Authors
In this manuscript the authors analyze the effect of native lectins from TBLF and a recombinant lectin (rTBL-1) on calcium mobility in breast cancer cells (MCF-7) and no-cancer MCF12F cells. They demonstrate that TBLF and rTBL-1 increased intracellular calcium concentration and mobilized calcium from intracellular stores in a dose-dependent manner.
The topic is interesting but there are some criticisms:
- the authors state that “normal cells and tumoral cells respond differentially to the lectins”. Why? The explanation that the authors give in the discussion is not exhaustive and further investigation would be necessary
- the authors state that “they we cannot distinguish the participation of calcium entry from the extracellular space or probable calcium release from the ER, the main calcium reservoir. To address this question, they conducted experiments under low extracellular calcium conditions. On the basis of obtained results they hypothesize a possible molecolar mechanism that involves specific second messanger. The manuscript would be more complete and useful if this mechanism were analyzed with specific experiments
- the title of the manuscript reflects the experiments performed in the manuscript but in my opinion they are only preliminary data and the topic should be explored further
- the antiproliferative effects of TFBL is evaluated both in normal and tumoral cells. In both cases the effect is greater at the higher TFLB concentration and from 12 h onwards it remains constant until 48 h. Please explain this point
Author Response
Comments and Suggestions for Authors
In this manuscript the authors analyze the effect of native lectins from TBLF and a recombinant lectin (rTBL-1) on calcium mobility in breast cancer cells (MCF-7) and no-cancer MCF12F cells. They demonstrate that TBLF and rTBL-1 increased intracellular calcium concentration and mobilized calcium from intracellular stores in a dose-dependent manner.
The topic is interesting but there are some criticisms:
- the authors state that “normal cells and tumoral cells respond differentially to the lectins”. Why? The explanation that the authors give in the discussion is not exhaustive and further investigation would be necessary
R: This is likely due to the fact that intracellular calcium regulation in MCF-12F cells is more efficient compared to MCF-7 cells. We believe this is a result of calcium rehostasis following the challenge by lectins (see lines 209-211). Additionally, our observations consistently show higher calcium levels in MCF-7 cells than in MCF-12F cells (data not shown; this will be presented in detail in our next paper).
- the authors state that “they we cannot distinguish the participation of calcium entry from the extracellular space or probable calcium release from the ER, the main calcium reservoir. To address this question, they conducted experiments under low extracellular calcium conditions. On the basis of obtained results they hypothesize a possible molecular mechanism that involves specific second messenger. The manuscript would be more complete and useful if this mechanism were analyzed with specific experiments
R: The effect we observe when applying TBLF and rLTB-1 to the cell lines is substantial. That is, calcium increases efficiently and rapidly floods the cytosol of a cell, generating an intercellular calcium wave. Based on this, it could even be discussed that this is an intracellular calcium wave activated by a calcium-induced calcium release event. As we mentioned previously, we are studying these events using specific drugs for the RyR and IP3 receptors, such as high concentrations of ryanodine and xestospongin to inhibit both receptors, respectively.
- the title of the manuscript reflects the experiments performed in the manuscript but in my opinion they are only preliminary data and the topic should be explored further
R: As we mentioned in the work, they are pioneers in the field. Our interest is in disseminating the observation that lectins are capable of mobilizing intracellular calcium in both cell lines, and that the mechanism of calcium homeostasis is what ultimately generates a differential effect between the two cell lines. We can argue that in our laboratory, we are conducting more specific experiments with drugs that help us differentiate the calcium release sites (RyR and IP3 receptors) to establish the biophysical and molecular mechanisms of how lectins act.
- the antiproliferative effects of TFBL is evaluated both in normal and tumoral cells. In both cases the effect is greater at the higher TFLB concentration and from 12 h onwards it remains constant until 48 h. Please explain this point
R. See lines 246 to 259.
Thank you very much for your observations.Round 2
Reviewer 1 Report
Comments and Suggestions for Authors
Figure 1 legend mentions a red and a black arrows, which were not included in the figure
Reviewer 2 Report
Comments and Suggestions for Authors
I thank the authors for their answers to my questions. Unfortunately, their answers are not convincing. The authors state that the aim of their work is to disseminate the observation that lectins are capable of mobilizing intracellular calcium in normal and tumoral cells, and that the mechanism of calcium homeostasis is what ultimately generates a differential effect between the two cell lines. But the experiments presented are not sufficient and further investigation is undoubtedly needed. The authors state that they will present more complete data in a subsequent work. This is very interesting but then this first work should be more complete, and not a simple preview of future more interesting results